# Human Endogenous Retrovirus, SARS-CoV-2, and HIV Promote PAH via Inflammation and Growth Stimulation

**DOI:** 10.3390/ijms24087472

**Published:** 2023-04-18

**Authors:** Desheng Wang, Marta T. Gomes, Yanfei Mo, Clare C. Prohaska, Lu Zhang, Sarvesh Chelvanambi, Matthias A. Clauss, Dongfang Zhang, Roberto F. Machado, Mingqi Gao, Yang Bai

**Affiliations:** 1Department of Clinical Pharmacology, School of Pharmacy, China Medical University, Shenyang 110122, China; 2Division of Pulmonary, Critical Care, Sleep, and Occupational Medicine, Department of Medicine, Indiana University, Indianapolis, IN 46202, USA; 3Department of Medicine, Brigham and Women’s Hospital, Harvard Medical School, Boston, MA 02115, USA; 4Department of Pharmacognosy, School of Pharmacy, China Medical University, Shenyang 110122, China

**Keywords:** human endogenous retrovirus K, pulmonary arterial hypertension, human immunodeficiency virus, SARS-CoV-2, inflammation

## Abstract

Pulmonary arterial hypertension (PAH) is a pulmonary vascular disease characterized by the progressive elevation of pulmonary arterial pressures. It is becoming increasingly apparent that inflammation contributes to the pathogenesis and progression of PAH. Several viruses are known to cause PAH, such as severe acute respiratory syndrome coronavirus-2 (SARS-CoV-2), human endogenous retrovirus K(HERV-K), and human immunodeficiency virus (HIV), in part due to acute and chronic inflammation. In this review, we discuss the connections between HERV-K, HIV, SARS-CoV-2, and PAH, to stimulate research regarding new therapeutic options and provide new targets for the treatment of the disease.

## 1. Introduction

Pulmonary arterial hypertension (PAH) is a pathophysiological syndrome characterized by a progressive increase in pulmonary vascular resistance and pulmonary arterial pressure, resulting in right heart dysfunction and ultimately leading to death [1]. The hallmarks of the disease include pulmonary vasoconstriction, the proliferation of pulmonary artery endothelial cells (PAECs), smooth muscle cells (PASMCs), the thickening of pulmonary vascular intima and adventitia, and peripheral inflammation [2]. The histopathological features of PAH are intimal hyperplasia, hypertrophy, and the epi membrane hyperplasia of arterioles with varying degrees of inflammatory reactions [3]. The exact pathophysiology of virus-associated PAH is currently unknown, but as with a limited understanding of the diseases, it is considered multifactorial. Human immunodeficiency virus (HIV), human endogenous retrovirus K (HERV-K), and severe acute respiratory syndrome coronavirus 2 (SARS-CoV-2) also lead to the release of proinflammatory factors, such as interleukin-6 (IL-6), IL-1β, and tumor necrosis factor α (TNF-α) [4,5,6,7]. Studies have shown that approximately 0.5% of HIV-infected individuals develop PAH, which is 100 to 1000 times higher than the prevalence of PAH in HIV-uninfected individuals [8]. As one of the most active members of the HERV genera, the upregulation of HERV-K can initiate and maintain immune system activation and contribute to PAH-related vascular changes, such as stimulating perivascular macrophages [6]. In addition, SARS-CoV-2 patients are at risk for cardiac and/or pulmonary complications due to pulmonary hypertension (PH) [9]. HIV, HERV-K, and SARS-CoV-2 may constitute a regulatory circuit. HERV-K not only affects the structure of HIV but also influences its infectivity primarily through the Envelope (Env) protein. HERV-K can regulate the expression of the HIV Gag protein, which affects the release of HIV virions [10]. SARS-CoV-2 can activate HERV-K [11] and HIV infection reduces immune cell function and increases the risk of acquiring SARS-CoV-2 [12]. In addition, factors such as hypoxia, complement system activation, hemodynamic stress, and viral protein-mediated ECs death may also contribute to the onset and progression of PAH during SARS-CoV-2 infection. Studies from New York have shown that any increased right ventricular load (hypoxemia, hypotension, etc.) in SARS-CoV-2 infection may lead to higher mortality in patients with more severe PH and right ventricular dysfunction [13]. Therefore, it is believed that HIV/HERV-K/SARS-CoV-2 may be involved in the development of PAH with a counterregulatory loop. In this review, we discuss the current understanding of the regulatory loops between HERV-K, HIV, SARS-CoV-2, and PAH. Our goal is to spur discussion exploring new directions for the treatment of PAH and provide new targets for the treatment of the disease.

## 2. HIV: A Cause of PAH

HIV is a member of the lentivirus genus, which includes retroviruses with complex genomes that exhibit conical capsid core particles [14]. Like all retroviruses, the genome of HIV is encoded by RNA, which is reverse transcribed into viral DNA by viral reverse transcriptase (RT) after entering new host cells [15]. Infection with HIV results in systemic T-cell destruction and reduced immunity [16]. HIV invades the human immune system, primarily through CD4+ T lymphocytes, monocytes, and macrophages. To enter the cell, the HIV protein Env binds to the primary cell receptor CD4+, which in turn binds to the cellular cognate receptor [17]. The hallmark of HIV infection is the depletion of CD4+T lymphocytes, eventually leading to defective cell-mediated immunity, which is significant enough and may lead to various opportunistic infections [18,19]. With the advent of modern antiretroviral therapy (ART) regimens, HIV-infected patients are suffering from non-AIDS-related comorbidities at a higher rate [20], including cardiopulmonary vascular diseases [21]. HIV infection is considered an independent risk factor for PAH [22]. The pathogenesis of HIV-associated PAH is complex but fundamentally driven by HIV-associated proteins [23,24] HIV-1 transmembrane glycol protein 120 (gp120) is a key protein responsible for HIV entry into target cells. Nef is expressed during the early stages of HIV infection and has been detected in the plexiform lesions of the SHIV Nef rhesus monkey model [25]. Nef plays a key role in HIV immune evasion by downregulating the expression level of CD4+ and disrupting the presentation of viral antigens by down-regulating type I major histocompatibility complex [26]. Moreover, the HIV-Nef protein persists in the lungs of HIV-Aviremic patients and induces EC death [27,28,29,30]. Nef can induce EC death after transferring from Nef-containing peripheral blood mononuclear cells to the endothelium via exosomes [29,30,31]. The ensuing impairment in endothelium-mediated vasodilation could act as an integral mediator in promoting PAH progression. In addition, some cardiopulmonary vascular regulators contribute to the progression of HIV-PAH. Endothelin-1 (ET-1), which is secreted by PAECs, can induce vascular smooth muscle cell (SMCs) proliferation, and ECs apoptosis, as well as activate macrophages [32,33,34,35]. Both increased ET-1 concentrations and inflammatory damage seems to promote apoptosis and cause PAH-related ECs growth and proliferation [36,37]. High levels of ET-1 are a risk factor for PAH [38] through dysregulated homeostasis [39,40,41]. Moreover, ET-1 promotes IL-6 production in macrophages [42] forming a paracrine amplification loop. Thus, in individuals with HIV, excessive ET-1 production can impair lung endothelial function and promote the development of PAH [41]. Studies have shown that the regulation of host gene transcription by HIV may mimic hereditary PAH by genetically altering the expression of bone morphogenetic protein receptor type 2 (BMPR2). The loss of BMPR2-dependent signaling is one of the main drivers of the pathogenesis of PAH [43,44]. IL-6 can negatively regulate the expression of BMPR2, causing the excessive proliferation of PASMCs [45,46]. Furthermore, it was found that Tat can inhibit the transcription of BMPR2 in macrophages, contributing to abnormal pulmonary vascular function and cell proliferation [47]. 

Moreover, the sterile alpha motif (SAM) domain and histidine-aspartate (HD) domain-containing protein 1 (SAMHD1) is part of the human immune system and is highly expressed in the lungs of patients with PAH [6]. It was found that SAMHD1 is a target antigen in the PAH lung immune complex and that elevated SAMHD1 in PAH lung cells and circulating classical dendritic cells [6]. Antibodies against SAMHD 1 are ubiquitous in tertiary lymphoid tissues stimulated by certain autoimmune diseases. SAMHD1 depletes cellular deoxynucleotide triphosphates (dNTPs) and prevents HIV-1 reverse transcription and inhibits HIV replication [48,49]. These associated mechanisms should be considered for treatment targets for patients with both PAH with HIV infection.

## 3. Role of SARS-CoV-2

SARS-CoV-2 is a recently discovered novel coronavirus that has caused significant morbidity and mortality and has been the focus of significant research since its emergence [50]. Its genome consists of 14 open reading frames (ORFs) arranged from 5′ to 3′: replicate (ORF1a/ORF1b), spike (S), envelope (E), membrane (M), and nucleocapsid (N). Among them, the spike (S) protein is a key surface glycoprotein that mediates the interaction between SARS-CoV-2 and its host cell receptors [51]. SARS-CoV-2 mainly attacks the angiotensin-converting enzyme 2 (ACE2) receptor expressed outside the airway epithelial cells, thereby enabling the rapid replication of SARS-CoV-2 in the lungs and causing respiratory disease [52].

### 3.1. SARS-CoV-2 and PAH

SARS-CoV-2 has been reported to cause cardiopulmonary complications [53], such as myocarditis, pulmonary embolism, as well as acute respiratory distress syndrome [54,55,56]. PAH patients have a high risk of rapid deterioration once infected with SARS-CoV-2 [57]. Suzuki et al. found pulmonary vascular wall thickening in the autopsies of patients with SARS-CoV-2, which is one of the hallmark signs of PAH [58]. Although research continues, it appears clear that PAH and SARS-CoV-2 shares pathological and physiological similarities. In the early stages of PAH, inflammatory cells can adhere to ECs and secrete proinflammatory mediators, leading to endothelial damage. At later stages, inflammatory cells secrete growth factors and chemokines that promote SMC migration, proliferation, and fibrosis [59]. The rapid replication of SARS-CoV-2 in the lung leads to the depletion of T lymphocytes, the inhibition of interferon signaling, and the production of many pro-inflammatory mediators and chemokines, resulting in the endothelial dysfunction and dysregulation of the immune system [60,61,62]. ACE-2 plays a major role in PAH and SARS-CoV-2, but it is unclear whether ACE-2 reduction in PAH will promote or suppress SARS-CoV-2-induced lung injury [63]. SARS-CoV-2 enters human cells by binding to ACE-2 receptors which are expressed in various lung cells, leading to the activation of known PAH pathways [64]. The reduction in ACE-2 in PAH may prevent SARS-CoV-2 viral entry into the lung [63]. Patients with PAH with severe hemodynamic features have been found to achieve good outcomes without heart failure [65]. This may be because the decreased ACE 2 expression in patients with PAH may play a protective role in the early stages of infection from viral entry [66], alternatively, PAH-associated vasodilator therapy limits the impact of virus-induced endothelial dysfunction [9,67]. Given the propensity of SARS-CoV-2 to infect the endothelium, it has also been proposed that the altered function of the endothelium and immune system in PAH may limit viral replication and suppress the harmful cytokine responses induced by SARS-CoV-2 [68,69]. These findings suggest that EC dysfunction and inflammation, a common feature of PAH and SARS-CoV-2, may lead to pulmonary vasoconstriction, thereby increasing pulmonary vascular resistance (PVR).

PAH may be a potential sequela of SARS-CoV-2 infection, particularly in patients with systemic hypertension [70,71]. Studies have shown that SARS-CoV-2 is active in promoting lung micro thrombosis, and vascular leakage, including inflammation, DNA damage, and mitochondrial dysfunction [72,73]. These observations in deceased patients infected with SARS-CoV-2, SARS-CoV-1, and H1N1 influenza viruses suggest that the thickening of pulmonary vascular walls is a unique feature of SARS-CoV-2 infection [74]. These changes may reflect the tissue response of the lungs to hypoxia, suggesting that patients recovering from SARS-CoV-2 infection may be susceptible to PH and right-sided heart failure [58]. Based on these studies, PAH can be considered a major risk factor for SARS-CoV-2 infection. Increased susceptibility to PAH may only occur in those who survive severe COVID-19, or in those who develop mild or no symptoms of SARS-CoV-2 infection. The results of an Italian study showed that the incidence of COVID-19 in patients with PAH was comparable to that of the general population but consistent with other chronic diseases and a higher risk of death [75]. Researchers reported a severe case of SARS-CoV-2 infection without the underlying conditions for PAH who developed PAH two months after being discharged [70]. Therefore, there may be many pathological compatibles between SARS-CoV-2 infection and PAH. Further evidence is needed to further elucidate the relationship between PAH and SARS-CoV-2.

Because PAH patients’ lung function has been impaired, their pulmonary blood vessels have become narrowed, and if they are infected with the SARS-CoV-2 virus, it may lead to a more serious condition. It is generally accepted that the COVID-19 prognosis for patients with PAH is determined by a combination of underlying PAH disease characteristics and risk stratification and other factors such as age, functional status, and comorbidities [65,76].

Over the past three years, several variants of SARS-CoV-2 have emerged, including alpha, beta, gamma, delta, and Omicron variants [77]. The role and mechanism of these variants of SARS-CoV-2 in PH are not fully understood but may contribute to pulmonary vascular remodeling, lung tissue inflammation, immune escape, and right ventricular hypertrophy. Further studies need to be conducted in the future.

In conclusion, although there are currently no large-scale studies on the relationship between SARS-CoV-2 infection and PAH, SARS-CoV-2 infection may negatively affect patients with PAH, worsening their symptoms of vascular lung disease.

### 3.2. SARS-CoV-2 and HIV

Studies have found a dysregulated inflammatory response in the lungs of patients with severe SARS-CoV-2, leading to a cytokine storm that may cause multiorgan damage [52]. The total number of CD4 + T cells decreases sharply during SARS-CoV-2 infection, leading to immune dysfunction in the body [61,78]. Interestingly, both HIV-1 and SARS-CoV-2 infection leaded to decreased CD4 T cell counts [79]. SARS-CoV-2 and HIV-coinfection have been found worldwide. The reduction in CD4+ T cell counts in HIV patients correlates with the severity of SARS-CoV-2 infection [80]. The proinflammatory milieu of HIV and SARS-CoV-2 is remarkably similar, which may increase the risk of SARS-CoV-2 infection in HIV patients. SARS-CoV-2 infection is associated with increased susceptibility to other respiratory infections, leading to urgent concerns about how to tailor therapy for HIV patients coinfected with SARS-CoV-2 [81].

## 4. HERV-K

HERV derives gene sequences that account for 8% of the human genome. It is believed that they were acquired through the multiple integrations of now-extinct exogenous retroviruses over the past 100 million years [82,83,84]. With an enhanced understanding of the functional sequences in the HERV genome and its interrelation with multiple diseases, HERVs have become a major focus of research. High levels of integrated HERV activate the immune system and are often associated with severe disease [85]. When exogenous retroviruses infect humans, they can integrate, in some cases permanently, into the human genome [86]. HERVs share high nucleotide similarity with exogenous retroviruses, and the interaction leads to the enhancement and restriction of exogenous retrovirus infection [87]. At the same time, there is substantial evidence that HERV protein expression is increased by proinflammatory cytokines [88,89]. Proinflammatory cytokines can induce HERV-K transcriptional activity [90] and HERVs may play an important role in inflammatory vascular disease. HERVs are roughly divided into three categories: Class I comprises γ-retro viruses, including HERV-H and HERV-W; Class II comprises β-retroviruses, including HERV-K; Class III comprises foamy viruses, such as HERV-L, HERV-S, and HERV-U [91,92]. HERV-K is the most active and intact group of endogenous retroviruses within the genome of primates [93,94]. HERV-K genome consists of four essential genes (*gag*, *pro*, *pol*, and *env*) flanked by long terminal repeat (LTR) sequences. The *gag* gene encodes viral core proteins such as the matrix, capsid, and nucleocapsid [95], the *pro* gene encodes protease involved in the viral life cycle, the *pol* gene encodes reverse transcriptase and integrase, and the *env* gene encodes envelope protein and accessory protein Rec or Np9 by alternative splicing caused by a 292-bp deletion at the boundary of the *pol* and *env* genes [96]. LTRs are organized as U3-R-U5 forward repeat units, and four functional domains are included: long-distance adjustment unit, enhancer unit, core promoter unit, and Tat response element. These domains regulate the expression levels of various viral structural and non-structural proteins [97] (Figure 1).

### 4.1. HERV-K and PAH

The upregulation of HERV-K can initiate and maintain the activation of the immune system and cause vascular changes such as PAH [6]. The deoxy uridine triphosphate nucleoside hydrolase (dUTPase) domain has been found in almost all HERV-K viruses [98]. PAH was induced in a rat model by experimentally using HERV-K dUTPase. These rats had a reduced pulmonary artery acceleration time, increased right ventricular systolic blood pressure, and right ventricular hypertrophy, demonstrating that HERV-K dUTPase can induce hemodynamic changes in PAH in an IL-6-independent manner [99]. Furthermore, dUTPase increased the sensitivity of PAECs to apoptosis [6]. HERV-K *env* and dUTPase RNA are elevated in PAH patients compared to unaffected individuals. HERV-K dUTPase can also induce hemodynamic and vascular changes in PAH and increase the sensitivity of PAECs to apoptosis in a manner independent from IL-6 [6]. In addition, HERV-K is expressed at high levels in CD68 + macrophages around blood vessels in PAH tissues. This induces a paracrine effect near vascular cells and B cells, further promoting a PAH-like phenotype in the vasculature [6]. Therefore, elevated HERV-K can promote PAH in patients. Interestingly, SAMHD1 is an innate immune factor that inhibits HIV replication, and its immune complexes are present in the lungs of PAH patients [6]. An elevated SAMHD1 causes the HERV-K gene product HERV-K envelope and dUTPase in PAH lungs to be elevated. In contrast, perivascular immune complexes containing the antiviral protein SAMHD1 are caused by elevated endogenous retroviral HERV-K products expressed in PAH perivascular macrophages and circulating monocytes. Therefore, HERV-K may co-regulate PAH with HIV.

### 4.2. SARS-CoV-2 Promotes HERV-K Activation

HERV-K expression is upregulated in SARS-CoV-2 infection and is closely associated with the dysregulation of the inflammatory response in vivo [11]. Chertow et al. showed that SARS-CoV-2 promotes HERV-K activation and that a high expression of HERV-K correlates with disease severity and early mortality [100]. Guo et al. also found that in the cyclic GMP–AMP synthase (cGAS) stimulator of the interferon genes (STING) (cGAS-STING) pathway, the high expression of HERV-K *gag*, *env*, and *pol* genes can induce the secretion of interferon in SARS-CoV-2 patients and the high expression of HERV-K increases with age or clinical classification. HERV-K may act as an endogenous regulator involved in the interaction between SARS-CoV-2 and TOST [101]. Therefore, studying HERV-K-related immune response mechanisms may help reduce the occurrence of inflammatory responses, which has important implications for the treatment of SARS-CoV-2.

### 4.3. The Relationship between HERV-K and HIV

In recent years, the relationship between HIV and HERV-K has been a subject of great interest [102]. Usually, most HERVs are epigenetically silenced or silenced by a mutation, however, under certain conditions, including irradiation, chemical exposures, or exogenous viral factors such as HIV, they may be activated [103]. Increasing the expression of HERV-K proteins may be related to AIDS-associated cancer [104]. However, endogenous retroviral peptides can directly regulate the body’s immune response. Anti-HERV-K cellular immune response in HIV-1-infected patients and HERV-K-specific T-cell clones eliminate HIV-1-infected cells in vitro [105,106,107]. This suggests that the detection of HERV viral proteins may affect innate immune system function [108,109]. Based on the variant of its envelope glycoprotein, HIV is subdivided into HIV-1 and HIV-2 [110]. The impact of HIV-1 infection on HERV-K is reflected in changes in mRNA expression levels, which then modify the expression of HERV-K *env* mRNA, resulting in the complete N-glycosylation of the HERV-K Env transmembrane units (TM) on the cell surface [111]. The progression of HIV-1 infection to AIDS is characterized by the increased presence of opportunistic infections as CD4+ counts decline [112]. Chronic viruses such as the Epstein–Barr virus induce the transcription of the *env* gene of HERV-K [113], suggesting that other viruses might also activate HERV transcription. Studies have speculated that the increased expression of HERV-K in PBMC in HIV-infected patients may be due to the indirect effects of HIV-1 infection [114]. It is also speculated that immune activation in HIV-1 infection may be an indirect mechanism leading to HERV-K expression [115]. The HIV-associated increase in HERV-K expression may be due to HIV-encoded proteins such as Tat and Vif that promote HERV-K gene expression. Both HIV-1 Tat and Vif are independent factors that can increase the expression of HERV-K *gag* RNA [114]. Only a few HERV-K proviruses are capable of independent replication [116], with studies demonstrating that Tat significantly activates the expression of multiple unique HERV-K proviruses [117]. HERV-K-specific CD8+ T cells obtained from human subjects eliminate HIV-1-infected cells in a Vif-dependent manner in vitro, resulting in the inhibition of viral replication [106]. Interestingly, HERV-K forms particles on the plasma membrane such as HIV-1 [118], and the formation of such particles is driven by HERV-K Gag. HERV-K gag consists of four domains: matrix (MA); capsid (CA); nucleocapsid (NC); p15; and short peptide sequences SP1, QP1, and QP2 [86]. Similarly, HIV-1 gag has four main domains: MA, CA, NC, and p6, with two spacer peptides SP1 and SP2 [119]. MA is necessary for HIV-1 Gag to target and bind PM. The NC domain can promote Gag multimerization during virus assembly [120]. The p6 domain contains the Pro-Thr-Ala-Pro (PTAP) domain, which recruits the cellular endosomal sorting complex required for transport (ESCRT) complexes to promote virion release. After undergoing viral particle release and viral Pro digestion, these domains produce individual mature Gag proteins. HERV-K Gag and HIV-1 Gag co-localize on the plasma membrane and can co-assemble into virions [121]. There is a competitive relationship between the two Gag proteins on the plasma membrane. HERV-K Gag will interfere with the binding of HIV-1 Gag to the plasma membrane and the release of HIV-1 alters the final permeability of the hetero multimer [121]. The assembly of HERV-K Gag reduces the HIV-1 released and interferes with the early stages of HIV-1 replication [120]. Monde et al. speculated that the interaction between HIV-1 Gag, and HERV-K Gag may interfere with the maturation of HIV-1 particles and the formation of viral cores [121]. In HIV-1-infected cells, HERV-K Gag is effectively assembled and processed into retrovirus-like particles under the mediation of viral Pro, indicating that the HIV protein expression can promote the release of HERV virus particles [122]. HIV-1 Gag can assist the release of HERV-K Gag virus particles, but HERV-K has no effect on HIV-1 Gag with the mutant PTAP that has virus release defects [121]. Although the mechanism by which HERV-K inhibits HIV-1 release is not well understood, HERV-K Gag certainly interferes with HIV-1 assembly through heterologous multimerization with HIV-1 Gag. In tissue culture models, it was shown that the overexpression of HERV-K Gag has been shown to reduce the release efficiency of HIV-1 and to impair HIV-1 infectivity [121]. Thus, it appears that HERV-K Gag can inhibit the formation of mature HIV-1 particles by co-assembly with HIV-1 Gag, regulate HIV-1 replication, and inhibit HIV-1 release efficiency and infectivity [120].

The constructed HERV-K consensus sequence (HERV-K con) can promote HIV infection by antagonizing the restriction factor tether [123]. In co-transfection experiments, plasmids encoding the functional Env protein of HERV-K con or HERV-K18 are used to assist in the assembly and release of HIV particles lacking Env [124]. However, there are functional differences between HERV-K Env sequences. Studies have shown that Env produced by endogenous proviruses K108 and K109 reduces HIV Gag protein levels and inhibits HIV release [10]. HERV-K108 is one of the best-studied HERV-Ks. Studies have shown that HERV-K108 Env consistently interferes with HIV-1 production [10] by inhibiting HIV-1 replication, but the protein encoded by the *env* sequence does not affect HIV-1 RNA metabolism. The furin Pro is a major protein invertase in the exogenous pathway that catalyzes the Arg-Xaa-Arg/Lys-Arg carboxy-terminal peptide bond (Xaa is any amino acid) in the original protein to produce a mature protein. Digesting viral glycoproteins is critical for viral particle maturation and infectivity [125]. Studies have shown that the inefficient action of HERV-K Env blocks HIV-1 replication, but the mechanism of action is unclear. It is known that this is not due to cytotoxicity, and the effect of HERV-K108 Env on HIV-1 is independent of cell type [10].

## 5. The Potential Mechanism of SARS-CoV-2/HERV-K/HIV in PAH

HIV and SARS-CoV-2 infection may both lead to a cytokine storm [126,127]. Meanwhile, HIV and SARS-CoV-2 infections are characterized by an increased incidence of thrombosis [128,129]. Current evidence suggests that like HIV, SARS-CoV-2 can invade ECs via ACE2, and stimulate the overproduction and release of inflammatory cytokines, which subsequently damage and activate ECs. These mechanisms appear to lead to the formation of emboli, leading to occlusion of the microvascular and pulmonary capillary network, contributing to the pathophysiology of acute respiratory distress syndrome (ARDS) in SARS-CoV-2 [128,130]. Interestingly, HIV-1 and SARS-CoV-2 infection can lead to the up-regulation of HERV-K, further leading to the increased synthesis of proinflammatory cytokines [11]. Both HIV-1 and HERV-K can promote the initiation and progression of inflammation. Inflammatory cells cause endothelial damage through the release of proinflammatory mediators, as well as secrete growth factors and chemokines. These promote the migration, proliferation, and fibrosis of fibroblasts and SMC, which can ultimately lead to the development of PAH. Thus, many pathological connections may exist between SARS-Cov-2, HERV-k, HIV, and PAH (Figure 2). These viruses may also affect the progression of PAH by affecting the release and migration of immune cells and inflammatory mediators (Table 1). We speculate that inflammation is a common feature of these diseases and that these inflammatory cells could be targeted for the treatment of coinfected patients.

## 6. Medical Therapy in PAH

Currently, therapies for patients with PAH include phosphodiesterase type 5 inhibitors, soluble guanylate cyclase stimulators, endothelin receptor antagonists, prostacyclin analogs, and prostacyclin receptor agonists [131]. Compared with single-agent targeted therapy, combination drug therapy is a better option for patients and can improve morbidity, although none influences mortality. HIV-related PAH treatments include sildenafil, bosentan, and prostacyclin analogs [132,133,134]. The coadministration of ritonavir and sildenafil has been successfully used in case reports [135]. According to the 2022 ESC/ERS Guidelines for the Diagnosis and Treatment of PAH, the treatment of HIV-PAH should follow the guidelines for the treatment of idiopathic PAH with HAART [136]. Currently, drug candidates for the treatment of SARS-CoV-2 patients include but are not limited to, redeliver, aziridine, dalbavancin, and pulmonary vasodilators [137,138,139,140]. It was recently discovered that soluble guanylate cyclase stimulators enhance the nitric oxide (NO) pathway as a classical pathway for the treatment of PAH, which may play a positive role in the treatment of SARS-CoV-2. Portable inhaled NO delivery systems are available for the treatment of patients with idiopathic PAH who developed COVID-19 pneumonia [141]. Additionally, clinical trials evaluating the potential benefit of inhaled NO in the treatment of COVID-19 pneumonia are ongoing [142].

## 7. Conclusions

We reviewed the available evidence suggesting that HERV-K, SARS-CoV-2, and HIV may play a regulatory role in the progression of PAH. HIV-1 infection causes the upregulation of HERV-K, which triggers and sustains the activation of the immune system and the development of inflammation, leading to the hallmark vascular changes seen in PAH. HERV-K affects the replication and release of HIV-1 through direct and indirect factors. SARS-CoV-2 activates HERV-K. SARS-CoV-2, HIV-1, and HERV-K can all promote the development and progression of inflammation, thereby causing endothelial damage and promoting the incidence and progression of PAH. Inflammation may be a common mechanism between these viruses and PAH, and similar inter-regulation may exist in other mechanisms, but further study is needed.

## Figures and Tables

**Figure 1 ijms-24-07472-f001:**
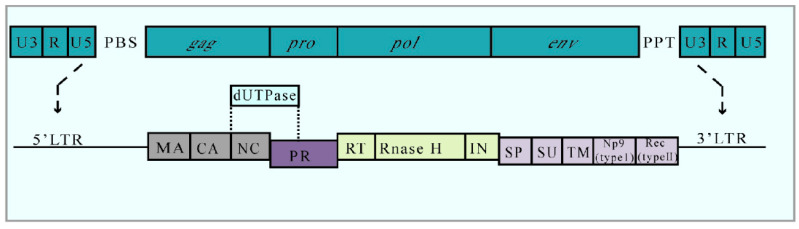
Block diagram of the typical HERV-K sequence. The consensus sequence of the HERV-K provirus has LTR, primer binding site (PBS), polypurine trait (PPT), and four main ORFs: (1) gag, which encodes structural proteins: matrix (MA), capsid (CA), and nucleocapsid (NC); (2) protease (pro), which also encodes the enzyme dUTPase; (3) polymerase (pol), which reverse transcriptase (RT), RNase H and integrase (IN) and (4) envelope (env), which has three different domains: the signal peptide (SP), surface (SU), and transmembrane (TM).

**Figure 2 ijms-24-07472-f002:**
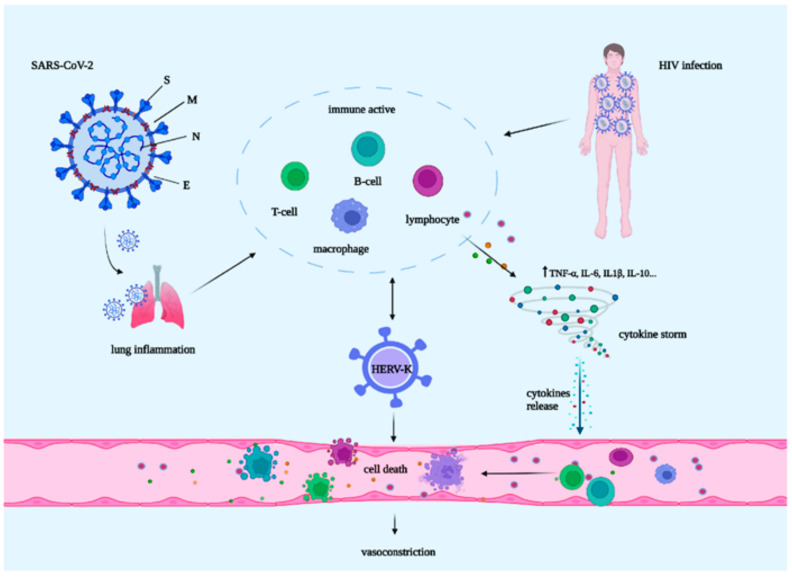
HERV-K, HIV, and SARS-CoV-2 may cause PAH by triggering inflammation. HIV invasion can lead to cellular immune deficiencies and enhance IL-6 secretion, which could increase the permeability of endothelial cells (ECs). Similarly, the invasion of SARS-Cov-2 into the lungs can lead to the dysregulation of the immune response, and the release of pro-inflammatory mediators such as IL-6, TNF-α, and other pro-inflammatory mediators causing endothelial damage. HIV and SARS-Cov-2 infections upregulate HERV-K protein expression, resulting in the increased synthesis of pro-inflammatory factors. Eventually, a storm of inflammatory factors formed through the immune response and the release of pro-inflammatory mediators can cause damage to ECs, and the migration, proliferation, and fibrosis of smooth muscle cells, thereby promoting the occurrence and development of PAH (picture created with biorender.com (accessed on20 September 2022)).

**Table 1 ijms-24-07472-t001:** Connection with virus and PAH.

Virus	Contributes to PAH
HIV	Tat + TNF-α → IL-6 → Endothelial permeability
Tat → BMPR2 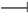 PASMC proliferation
ET-1 → IL-6 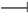 BMPR2 → PASMC proliferation
HERV-K	dUTPase → Sensitivity of PAEC to apoptosis
SARS-CoV-2	Depletion of T lymphocyte inflammatory responses 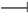 endothelial function

## Data Availability

All data are available in the manuscript and figures.

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
