# Peer review of "Human Endogenous Retrovirus, SARS-CoV-2, and HIV Promote PAH via Inflammation and Growth Stimulation"

_ijms, 2023, doi:10.3390/ijms24087472_

Round 1

Reviewer 1 Report

The manuscript by Wang et al. addressed the association between HERV-K, HIV, SARS- 19 CoV-2, and PAH. However, the information provided by the authors is limited and the major conclusions are confusing/misleading. In addition, the central issue of the manuscript is not clearly defined. It seems that the authors attempt to link HERV-K to HIV- and SARS-19 CoV-2-assocaited PAH. However, they failed to reach a clear conclusion. My major concern is that the manuscript lacks novel insights or opinions on the mechanisms and management of virus-induced PAH, especially for SARS-19 CoV-2-associated PAH. Keep in mind that in addition to dysregulated inflammation, factors such as hypoxia, activation of the complement system, hemodynamic stresses, and viral proteins-mediated endothelial cell death, may contribute to the initiation and progression of PAH during SARS-19 CoV-2 infection. Overall, I have serious concerns about the suitability of this manuscript for publication.

Author Response

Thank you for your recommendations. We revised the manuscript. Please see the attachment.

Reviewer 2 Report

In this review, the authors discussed the current understanding of the regulatory loops between several viruses’ infection and pulmonary arterial hypertension. The manuscript is well written and most of the points clearly presented. Moreover, the topic of this review is important in the PAH field, especially these related to SARS-CoV-2. I recommend this manuscript be accepted after the authors addressed the following questions.

1.   1. It has been suggested that PAH could be a potential sequela of COVID-19 in particular in those with systematic hypertension. Please discuss the role of SARS-CoV-2 infection in PAH patients and the possible mechanism underlying it. is the PAH patients more prone to of SARS-CoV-2 infection? Is there any difference in the prognosis of COVID-19 in PAH patients?

2. 2. If possible, please discuss the difference of SARS-CoV-2  variants in regulating the PAH-related pathological features.

Author Response

(The authors gave the same response as above.)

Reviewer 3 Report

The authors have reviewed the current literature on the development of pulmonary arterial hypertension (PAH) and its association with viral infection using examples of HERV-K, HIV,  and SARS-CoV-2. In this concise review, the authors have provided general information without going deeply into the published literature. I have the following comments.

1.     It is a general comment, the review does provide all the valuable information but there is a lack of clarity in the writing. The transitioning from one topic to another within the paragraph is very abrupt and sometimes it really doesn’t connect. For example, on page 3, line 104, The authors described the role of BMPR2 in the pathogenesis of PAH and then suddenly switched to SAMHD 1 abruptly.  Also, the role of SAMHD 1 and its association with PAH is not described at all. The author needs to describe the study.

2.     In the conclusion the authors have indicated a very strong role of HERV-K in the pathogenesis of PAH by associating it with SARS-COV2 and HIV infection. However, the authors have written a very small section with only two papers about the HERV-K and PAH link. That section is not very well written, it is not clear whether HERV-K can only initiate PAH in patients with HIV and SARS-CoV-2 or can do so alone. Please clarify the text.

3.     Page 2, lines 77-79. The sentence doesn’t make sense, please correct it.

4.     Page 2, lines 85-86, please clarify the sentence.

Author Response

Thank you for the recommendations. we addressed the manuscript under your suggestions. Check the details below, please.

Point 1: It is a general comment, the review does provide all the valuable information but there is a lack of clarity in the writing. The transitioning from one topic to another within the paragraph is very abrupt and sometimes it really doesn’t connect. For example, on page 3, line 104, The authors described the role of BMPR2 in the pathogenesis of PAH and then suddenly switched to SAMHD 1 abruptly.  Also, the role of SAMHD 1 and its association with PAH is not described at all. The author needs to describe the study.

Response 1:Thank you very much for your review comments. We illustrate the connection between SAMHD 1 and PAH in our revised manuscript (Please see lines 109-111).

Point 2: In the conclusion the authors have indicated a very strong role of HERV-K in the pathogenesis of PAH by associating it with SARS-COV2 and HIV infection. However, the authors have written a very small section with only two papers about the HERV-K and PAH link. That section is not very well written, it is not clear whether HERV-K can only initiate PAH in patients with HIV and SARS-CoV-2 or can do so alone. Please clarify the text.

Response 2: Thank you very much for reviewing and commenting on our paper. We clarify conclusions regarding the relationship between HERV-K and PAH in this section (Please see lines 244-252).

Point 3: Page 2, lines 77-79. The sentence doesn’t make sense, please correct it.

Response 3: Thank you for your valuable suggestions on our paper. We have deleted this meaningless sentence.

Point 4: Page 2, lines 85-86, please clarify the sentence.

Response 4: Thank you for your valuable suggestions on our paper. We have revised the sentence in the revised version. 

Round 2

Reviewer 2 Report

My concerns have been addressed.

Author Response

Thank you for your review and suggestions. 

Appreciate.

Reviewer 3 Report

My concerns are addressed by the reviewer. I have no more comments.